Proj2Proj: self-supervised low-dose CT reconstruction

Unal Mehmet Ozan 1 unalmehmet@itu.edu.tr
http://orcid.org/0000-0003-4031-4656 Ertas Metin 2
Yildirim Isa 1
1 Department of Electronics and Communication Engineering, Istanbul Technical University , Istanbul , Turkey
2 Department of Electrical and Electronics Engineering, Istanbul University-Cerrahpasa , Istanbul , Turkey
Chaki Jyotismita
Electronic publication date: 2024 Feb 29
Publication date: 2024
Volume: 10
Electronic Location ID: e1849
Received 2023 Jul 13; Accepted 2024 Jan 10
Copyright: © 2024 Unal et al.
Copyright year: 2024
Copyright holder: Unal et al.
License: This is an open access article distributed under the terms of the Creative Commons Attribution License, which permits unrestricted use, distribution, reproduction and adaptation in any medium and for any purpose provided that it is properly attributed. For attribution, the original author(s), title, publication source (PeerJ Computer Science) and either DOI or URL of the article must be cited.
License URL: https://creativecommons.org/licenses/by/4.0/

Keywords: Deep learning, Low-dose CT, Image reconstruction, Self-supervised learning

Funding: The authors received no funding for this work.

==============================
In Computed Tomography (CT) imaging, one of the most serious concerns has always been ionizing radiation. Several approaches have been proposed to reduce the dose level without compromising the image quality. With the emergence of deep learning, thanks to the increasing availability of computational power and huge datasets, data-driven methods have recently received a lot of attention. Deep learning based methods have also been applied in various ways to address the low-dose CT reconstruction problem. However, the success of these methods largely depends on the availability of labeled data. On the other hand, recent studies showed that training can be done successfully without the need for labeled datasets. In this study, a training scheme was defined to use low-dose projections as their own training targets. The self-supervision principle was applied in the projection domain. The parameters of a denoiser neural network were optimized through self-supervised training. It was shown that our method outperformed both traditional and compressed sensing-based iterative methods, and deep learning based unsupervised methods, in the reconstruction of analytic CT phantoms and human CT images in low-dose CT imaging. Our method’s reconstruction quality is also comparable to a well-known supervised method.

Introduction

Computed Tomography (CT) imaging is one of the most common tools used in the diagnosis of diseases. The inside of the human body can be monitored and problematic tissues, deformities, and lesions can be detected via CT. Despite these benefits, CT imaging has one crucial downside: ionizing radiation. The radiation dose reduction in CT imaging is possible in several ways such as: i) decreasing the number of projections, ii) reducing of X-ray tube current, iii) narrowing the viewing angle.

During CT imaging, the source emits X-ray, and the detector data at different angles are collected. These gathered data are known as CT projections, and the image created from them is known as a CT image. The reconstruction in CT is the creation of images from these noisy indirect projections. Filtered back projection (FBP), the most traditional method used in CT imaging, works by projecting CT projections back to the image domain. With the help of a filter, since low frequencies are sampled more than high frequencies due to the system geometry, it reduces the effect of low frequencies and increases the weight of high frequencies. However, this operation increases the sensitivity of the FBP method to noise. Therefore, when the projections are noisy or incomplete, FBP method does not produce satisfactory results.

To overcome this limitation of the FBP method, iterative methods were developed. Simultaneous algebraic reconstruction technique (SART), one of the most popular of these methods, simultaneously back-projects the error for all projections (Andersen & Kak, 1984). Recently compress sensing based iterative reconstruction methods have been proposed. Compressed sensing states that: if the information is sparse on a known basis, this information can most likely be recovered by incomplete measurements (Donoho, 2006; Candès, Romberg & Tao, 2006). Given that the information to be recovered is not sparse in its original space, sparsifying transforms can be used. Among them, total variation (TV) has been extensively used. TV minimization method aims to minimize the gradient magnitude of the images, which constrains the solution set to piece-wise smooth images (Rudin, Osher & Fatemi, 1992). Iterative and compressed-sensing-based methods were merged and proposed for low-dose CT problem (Yu & Wang, 2009; Sidky & Pan, 2008). Image domain denoisers were also used to address this problem. Non-Local means (Buades, Coll & Morel, 2005) and Block Matching 3D (BM3D) (Dabov et al., 2007) exploit the non-local similarities of the images and use these similarities for denoising. These methods are quite hyperparameter dependent and use simplistic hand-crafted priors.

Nowadays, deep learning based studies have become attractive with the increase in computing capacity and the availability of large datasets. Numerous deep learning based methods have been proposed for image domain inverse problems such as denoising, deconvolution, and inpainting. Deep learning based methods have also been applied to the low-dose CT problem. Convolutional neural networks (CNN) based methods were used as denoisers in the image domain (Jin et al., 2017; Chen et al., 2017; Buzug, 2008; Yang et al., 2018; Liu et al., 2020). Deep learning methods were also extended by applying iteratively with classical reconstruction techniques (Adler & Öktem, 2018; Wu et al., 2017; He et al., 2019).

Advancements in the deep learning field made it possible to define sophisticated learning methods and learned constraints to further optimize low-dose CT reconstructions. Wu et al. (2021) applied a residual-based network method by optimizing the network using both measurement consistency and image quality awareness. Spectral2Spectral (Guo et al., 2023) study used the similarity prior within the image-spectral domain as a regularization term to constrain the network training. Deep learning methods were also used in metal artifact reduction by applying the loss in the dual domain (Zhou et al., 2022). Another method to improve the image quality of low-dose CT images is to denoise the projections. Yang et al. (2023) applied this method by using transformer architecture with self-attention. Wu et al. (2023) proposed a wavelet-improved denoising for low-dose CT problems that exploit the score-based generative model approach.

For most of the deep learning based methods given, one of the most important factors of the success is the availability of large datasets. Another field of focus aims to tackle this problem by enabling denoising without the need for noisy-denoised pairs datasets. In order to address this issue, deep image prior (DIP) study suggested to use the architecture of CNNs as a regularizer. Baguer, Leuschner & Schmidt (2020) combined this approach with TV regularizer and applied this unsupervised deep CNN based method to low-dose CT problem. Noise2Noise proposes a method for the denoising problem that does not require noise-free target images (Lehtinen et al., 2018). However, it still requires two independent noisy measurements of the same information. Noise2Self (Batson & Royer, 2019) and Noise2Void (Krull, Buchholz & Jug, 2019) studies, on the other hand, suggested that denoising can be performed using only the noisy measurement itself. In other words, the image itself is used as the target image during the model training. One of the problems that may arise at this point is that the model can converge to an identity function. Noise2Self study suggested a kind of perturbation mechanism called Jth invariant principle. Noise2Void study approached the denoising problem on a pixel scale and defined a receptive field so that the pixel was not used during the estimation of that pixel. Subsequently, studies (Quan et al., 2020; Xie, Wang & Ji, 2020) were also proposed to prevent this convergence in different ways.

The opportunity of training on noisy target images could be quite valuable for low-dose CT reconstruction. Since it is not always easy to obtain low-dose/normal-dose pairs to create big datasets, various learning methods without noise-free targets have been proposed. Noise2Inverse (Hendriksen, Pelt & Batenburg, 2020) study proposed a learning method that exploits Noise2Noise (Lehtinen et al., 2018) principle by grouping projections and using them as targets against each other. To learn a reconstruction method from a single image, Noise2Filter (Lagerwerf et al., 2020) study combined Noise2Inverse and NN-FBP (Pelt & Batenburg, 2013) methods.

In our study, we propose a method so-called Proj2Proj which customizes Noise2Self method for the low-dose CT reconstruction problem by applying the self-supervision principle in the projection domain. We used the self-supervision to train a neural network that denoises the reconstruction. We were able to train this neural network with only low-dose projections without the need for low-dose/normal-dose pairs whose availability might be a big concern in this field. Our method outperformed traditional and compressed sensing-based iterative methods and an unsupervised deep CNN based method in the reconstruction of analytic phantoms and human CT images both qualitatively and quantitatively. Furthermore, it produced such results that were comparable with a well-known supervised approach, FBP+U-Net (Jin et al., 2017). In addition to peak signal-to-noise (PSNR) and Structural Similarity (SSIM), the Learned Perceptual Image Patch Similarity (LPIPS) (Zhang et al., 2018) metric was also used for the qualitative comparisons, which focuses on the perceptual similarity of the results. The preliminary results of this study were published in a leading conference in biomedical imaging. The present article extends (Unal, Ertas & Yildirim, 2021) by detailing the mathematical definition of the method and adding theoretical justification of the method, by training the network with two different datasets, by extending the experiments with self-supervised and supervised deep learning based methods and new metrics.

The article is organized as follows. The rationale and implementation of the self-supervised training method are described in “Method”. The experiment datasets and settings, as well as the findings, are presented in “Experiments”. “Discussion” discusses further details of the method. Finally, the last section brings the article to a conclusion.

Method

Normal-dose CT image acquisition process can be defined as follows:

(1) y𝖭=A𝖭x∗+η,x𝖭=𝖥𝖡𝖯(y𝖭)

where x∗ is the ground-truth X-ray attenuation of the body, A𝖭 is the normal-dose data acquisition matrix, η is the data acquisition noise, y𝖭 is the normal-dose CT projections. Conventionally, the normal dose CT image is reconstructed with the 𝖥𝖡𝖯 method.

The low-dose reconstruction process can be represented as:

(2) y𝖫=A𝖫x∗+η,x𝖫=𝖥𝖡𝖯(y𝖫)

where x∗ is the ground-truth image, A𝖫 is the low-dose data acquisition matrix, y𝖫 is the low-dose CT projections. Similarly, the low-dose CT image can be reconstructed with 𝖥𝖡𝖯 method and represented here as x𝖫. However, x𝖫 does not usually have sufficient image quality when it is compared with the ground truth image x∗.

In the presence of normal-dose CT projections, a supervised reconstruction model can be shown with the following equation:

(3) θ∗=arg⁡minθ𝔼x[||fθ(x𝖫)−x𝖭||22]

where fθ is the deep neural network (DNN) which is parameterized with θ, 𝔼x is the expectation over different x images, x𝖫 is the reconstructed low-dose image defined in (2), and x𝖭 is the reconstructed normal-dose image defined in (1). The supervised model given in (3) can be used in projection domain as follows:

(4) θ∗=arg⁡minθ𝔼y[||A𝖭fθ(x𝖫)−y𝖭||22]

where 𝔼y is the expectation over different y projections. For the supervised solution to the low-dose CT problem, both low-dose and normal-dose pairs of the same information are required. DNN which is defined with fθ learns how to map a low-dose image to a normal-dose one. However, it is not easy to collect such data due to the sensitivity of human tissues to ionizing radiation. The proposed method requires only the low-dose projections y𝖫 without the need of y𝖭 or x𝖭. This idea was formulated by defining a self-supervised loss that uses only low-dose projections and images, y𝖫 and x𝖫.

(5) 𝔼x,y[||A𝖫fθ(x𝖫)−y𝖫||22]

where 𝔼x,y is an expectation over x and y random variables and the rest of the variables are defined in (1) and (2). Our proposed self-supervised loss can be proven to converge the supervised solution. Let’s replace y𝖫 in (5) with A𝖫x∗+η using (2):

(6) 𝔼x,𝖭[||A𝖫fθ(x𝖫)−A𝖫x∗−η||22]

If (6) is rewritten by expanding the l2-norm square:

(7) 𝔼x[||(A𝖫fθ(x𝖫)−A𝖫x∗)||22]+𝔼η[||η||22]−𝔼x,η[2⟨A𝖫fθ(x𝖫)−A𝖫x∗,η⟩]

The second term is the expectation over η of ||η||22 which is the variance of the noise and not taken into account during the minimization since its gradient equals to zero. The third term of the equation is expectation over x,η of −𝔼x,η[2⟨A𝖫(fθ(x𝖫)−x∗),η⟩] whose weight becomes minimal in (7) since fθ(x𝖫)−x∗ ideally converges to zero. As a result, the first term, 𝔼x[||(A𝖫fθ(x𝖫)−A𝖫x∗)||22], becomes the most effective term in this optimization problem which is actually correlated with the supervised loss defined in (3) and uses x∗ random variable as the target.

In the previous paragraph, it is shown why self-supervised loss can be an effective candidate to learn a mapping from low-dose to normal-dose. However, fθ might converge to an identity function in (5). To overcome this issue, Jth invariant principle was applied (Batson & Royer, 2019) which proposes that N subset of the pixels are selected from noisy target images and the selected pixels are perturbed using the neighbor pixels, and the loss is defined based on only these pixels as in (8). In this way, the convergence of fθ to an identity function is prevented. To apply this principle to our Proj2Proj loss function, (5) is rewritten as follows:

(8) θ∗=arg⁡minθ𝔼y[||J(A𝖫fθ(𝖥𝖡𝖯(y𝖫Jc))−y𝖫)||22]

where J is the mask which filters only the Jth subset of the pixels, y𝖫Jc is the low-dose projections whose Jth subset of pixels are perturbed. Finally, the proposed loss function can learn from only low-dose projections without the need of normal-dose images or projections.

The complete working schema of the proposed Proj2Proj reconstruction method is also given in Fig. 1. The process can be examined in three parts as preprocessing, training, and reconstruction and evaluation. In preprocessing, low-dose projections are created from the CT images via low-dose CT forward operator. The obtained projection dataset is split into training, validation, and test sets. For the training set projections, the Jth subset of the pixels is perturbed and given as the input to FBP reconstruction algorithm. The perturbations are done via the following steps: The input image is divided into square pieces with the help of a grid. In our setup, it is selected as 4×4.

The pixel ( ith) to be perturbed is selected by taking the modulo of the number of iterations over the number of pixels in the grid (in our case 4 × 4 = 16). For example, if the iteration number is 17, 17mod(16) equals to 1.

ith pixel of all the grids are perturbed with the average of all four neighbor pixels excluding the ith pixel itself.

Figure 1 Proposed working schema for self-supervised low-dose CT reconstruction.

Image source credit: Yang et al. (2018).

The output of FBP is denoised with the neural network fθ and forward projected to projection space via low-dose CT forward transform. The error is calculated between the output and the perturbed projections after the same Jth subset mask is applied to the error. The neural network is optimized by minimizing the loss function in (8) with the Adam (Kingma & Ba, 2015) optimizer. After the network weights ( θ) are optimized by the training, they are used for the reconstruction. In the reconstruction phase, the input projections are given to the FBP operator in raw form without perturbation. The output of the FBP operator is denoised with the learned neural network fθ∗ whose output is the reconstructed image by the proposed approach.

Experiments

The details of experiments and code are given at the code repository (https://github.com/mozanunal/SparseCT).

Experimental setup

Deep lesion dataset (Yan et al., 2017) as human CT data and Shepp-Logan phantom and ellipses dataset as synthetic data were used in the experiments. The image resolution was selected as 512 × 512. The ellipses dataset consists of 36,400 artificially generated images and it was split into training (32,000 slices), validation (3,200 slices), and test (3,200 slices) datasets. DeepLesion dataset consists of 32,120 CT slices from 10,594 studies of 4,427 unique patients. It was split into three datasets as training (60%), validation (20%), and test (20%). For simulated low-dose CT setup, 64-view parallel beam projections were generated using Radon transform, and detector resolution was selected as 512 × 1 pixels. The projections were contaminated with the noise level between 30–40 dB.

Comparison methods

FBP, SART (Andersen & Kak, 1984), SART+TV (Yu & Wang, 2009), SART+BM3D (Dabov et al., 2007), DIP+TV (Baguer, Leuschner & Schmidt, 2020) and FBP+U-Net (Jin et al., 2017) methods were compared with our self-supervised reconstruction method. Inverse Radon transform was used with a ramp filter for FBP. Iteration number = 40 and relaxation parameter = 0.15 were chosen for SART hyperparameters. The TV weight was set to 0.9 and sigma parameter of BM3D was chosen as 0.35. All these hyperparameters were chosen to generate the highest PSNR values in the validation sets. In addition to these methods, another BM3D (sigma = 0.20) reconstructor was used in the comparisons which gives the highest SSIM (Wang, Simoncelli & Bovik, 2003) for the validation set. For the implementation of the FBP+U-Net, the same network architecture of the proposed method was used and the number of parameters and iteration number were selected as the same for a fair comparison. For the realization of DIP+TV method, the reference study was used (Baguer, Leuschner & Schmidt, 2020) and Skipnet (Wang et al., 2018) was used as the network architecture. The network overfitted to a single image for 4,000 iterations with a learning rate of 0.02. The hyperparameters were obtained from the validation dataset.

Network architecture and training

U-Net, a commonly used architecture in denoising applications, was selected for the denoiser neural network. Since U-Net is an auto-encoder, it is successful at extracting features of the images and then decoding them from these latent representations (Chen et al., 2017). Besides, thanks to the skip connections between the scales of the encoder and the decoder, the gradient flow is facilitated and the higher-order structure can be transferred to the output more easily. In our model, a five-scale structure was preferred with the convolutional filter numbers being 64, 128, 256, 512, 1,024 from the first layer to the last layer.

The number of parameters and training iterations were chosen as 2.160.000 and 200.000, respectively. In total, the self-supervised method was trained for 28 hours on a computer with an RTX2080 TI graphics card. The learning rate was chosen as 0.0001 and ADAM (Kingma & Ba, 2015) optimizer was used.

The proposed self-supervised method was trained with two different schemes: i) Proj2Proj method trained on ellipses dataset, ii) Proj2Proj method trained on human CT dataset. Each was used in the comparisons of the corresponding dataset.

Results

In order to validate the proposed method, the experiments were done on Shepp-Logan, ellipses dataset and human CT dataset with different noise levels. PSNR, SSIM, and LPIPS metrics were used for quantitative assessment. In addition to the numerical analysis, 1-D profile analysis and visual examination for qualitative assessment were used.

Table 1 summarizes the quantitative performance of reconstruction methods with 64-view CT projections from the ellipses dataset at different noise levels. FBP produced the poorest results based on these metrics for all noise levels. The performance of SART was significantly improved when its output image was denoised via TV or BM3D with different σ values. As an unsupervised deep CNN based method, DIP+TV, showed slightly worse results than iterative and regularized reconstruction methods in general. However, the performance of DIP+TV increases with decreasing noise level. Considering the unsupervised architecture, Proj2Proj outperforms DIP+TV in all metrics, especially in higher noise levels which is the main challenge in low-dose CT reconstruction. However, the supervised method generated significantly better results than Proj2Proj in terms of PSNR and SSIM values. LPIPS aims to mimic human perception in terms of measuring image similarity. Therefore, it is a good candidate for measuring the quality of CT images in terms of clinical usability. For the LPIPS metric, the proposed Proj2Proj method produced significantly higher results than iterative and regularized methods and a competitor unsupervised method, DIP+TV. Moreover, it produced slightly higher values than the supervised equivalent of our method, FBP+U-Net and this is also supported by the visual assessments.

Table 1 The ellipses dataset results for 64 projections with the noise level 30 to 40 dB SNR.

	30 dB SNR	33 dB SNR	37 dB SNR	40 dB SNR	
PSNR	SSIM	LPIPS	PSNR	SSIM	LPIPS	PSNR	SSIM	LPIPS	PSNR	SSIM	LPIPS	
FBP	12.14±1.66	0.25±0.01	0.23±0.03	14.97±1.62	0.29±0.02	0.28±0.02	18.52±1.54	0.38±0.03	0.37±0.01	20.84±1.55	0.46±0.04	0.39±0.02	
SART	19.38±1.62	0.41±0.04	0.50±0.05	21.76±1.56	0.50±0.05	0.51±0.04	24.36±1.61	0.63±0.05	0.55±0.06	25.77±1.69	0.71±0.05	0.61±0.06	
SART+TV	27.59±1.93	0.85±0.03	0.60±0.02	27.84±1.95	0.87±0.03	0.68±0.07	27.99±1.97	0.88±0.02	0.73±0.05	28.04±1.97	0.88±0.02	0.81±0.02	
BM3D 0.35	27.95±2.08	0.90±0.03	0.60±0.09	28.45±2.01	0.91±0.02	0.68±0.08	28.70±1.94	0.92±0.02	0.79±0.04	28.76±1.94	0.92±0.02	0.84±0.04	
BM3D 0.20	26.89±2.26	0.85±0.07	0.62±0.07	28.27±2.13	0.90±0.03	0.71±0.07	28.76±2.03	0.92±0.02	0.77±0.05	28.87±2.06	0.92±0.02	0.86±0.03	
DIP+TV	25.84±2.21	0.79±0.08	0.50±0.12	27.36±1.83	0.85±0.05	0.65±0.07	28.55±1.78	0.90±0.03	0.79±0.06	28.82±1.76	0.91±0.02	0.89±0.02	
FBP+U-Net	30.48±1.80	0.93±0.02	0.81±0.05	31.27±1.79	0.94±0.01	0.83±0.05	31.90±1.85	0.95±0.01	0.86±0.03	32.21±1.89	0.95±0.01	0.90±0.01	
Proj2Proj	28.12±1.70	0.91±0.02	0.81±0.04	28.36±1.76	0.92±0.01	0.85±0.02	28.54±1.85	0.92±0.01	0.87±0.02	28.63±1.85	0.93±0.01	0.92±0.01	

The reconstruction of Shepp–Logan phantom (Shepp & Logan, 1974) results are given in Fig. 2. FBP reconstruction suffered from severe artifacts. SART generated better results than FBP however it still contains significant background noise. Although TV and BM3D suppress this background noise successfully to some extent, the detectability of fine details was adversely affected due to their over-smoothing effect. Compared to these methods, Proj2Proj method performed the best in both suppressing the background noise and recovering the fine details. Considering deep CNN based reconstructions, DIP+TV suffers from deformities in details though it provides a smoother background than SART. However, Proj2Proj and supervised FBP+U-net produced better results than all other methods with a smoother background in Proj2Proj. For a better qualitative comparison, Fig. 2 is zoomed which favored our method in recovering sharper features. In Fig. 3, the reconstruction results of an ellipses image with a noise level of 33 dB are given. When the performance of all methods are compared, there is a strong analogy with those obtained for Shepp–Logan phantom given in Fig. 2. Our proposed Proj2Proj method gave superior results in both suppressing background noise and recovering the morphological structures.

Figure 2 Shepp–Logan phantom reconstruction results from 64-view projections with 37 dB noise level: (A) ground truth, (B) FBP, (C) SART, (D) SART+TV, (E) SART+BM3D ( σ=0.35), (F) SART+BM3D ( σ=0.20), (G) DIP+TV, (H) FBP+U-Net, (I) Proj2Proj trained on ellipses dataset.

Figure 3 Ellipses image reconstruction results from 64-view projections with 33 dB noise level: (A) ground truth, (B) FBP, (C) SART, (D) SART+TV, (E) SART+BM3D ( σ=0.35), (F) SART+BM3D ( σ=0.20), (G) DIP+TV, (H) FBP+U-Net, (I) Proj2Proj trained on ellipses dataset.

Image source credit: Ellipses dataset, https://github.com/jleuschn/dival/tree/master/dival/datasets.

Table 2 summarizes the quantitative performance of reconstruction methods with 64-view CT projections from human CT dataset at different noise levels. Proj2Proj method gave better performance than the state-of-the-art regularized iterative methods and DIP+TV. Since FBP+U-Net was trained in a supervised setting on the human CT data, it produced higher metrics in terms of the PSNR and SSIM. It should be emphasized that these metrics totally rely on the perfectness of ground truth images. However, in terms of LPIPS values, Proj2Proj produced similar results with FBP+U-Net.

Table 2 The human CT dataset results for 64 projections with the noise level 30 to 40 dB SNR.

	30 dB SNR	33 dB SNR	37 dB SNR	40 dB SNR	
PSNR	SSIM	LPIPS	PSNR	SSIM	LPIPS	PSNR	SSIM	LPIPS	PSNR	SSIM	LPIPS	
FBP	15.03±1.16	0.30±0.02	0.27±0.02	17.51±0.95	0.36±0.02	0.28±0.02	20.28±0.63	0.44±0.02	0.31±0.01	21.84±0.57	0.50±0.02	0.34±0.01	
SART	22.20±1.03	0.51±0.04	0.67±0.09	24.29±0.79	0.62±0.04	0.68±0.08	26.41±0.69	0.73±0.03	0.69±0.04	27.44±0.78	0.78±0.03	0.71±0.05	
SART+TV	25.95±0.80	0.84±0.01	0.66±0.07	26.02±0.81	0.85±0.01	0.72±0.05	26.04±0.82	0.86±0.01	0.79±0.03	26.05±0.82	0.87±0.01	0.84±0.02	
BM3D 0.35	28.29±0.93	0.92±0.01	0.75±0.06	28.38±0.96	0.92±0.01	0.79±0.05	28.44±0.98	0.93±0.01	0.82±0.04	28.45±0.99	0.93±0.01	0.86±0.02	
BM3D 0.20	28.80±0.96	0.91±0.01	0.75±0.06	29.13±0.98	0.93±0.01	0.79±0.05	29.25±1.02	0.93±0.01	0.82±0.04	29.27±1.03	0.93±0.01	0.86±0.02	
DIP+TV	25.49±0.69	0.86±0.03	0.79±0.05	26.14±0.84	0.89±0.02	0.81±0.04	26.40±0.89	0.91±0.01	0.84±0.04	26.45±0.91	0.91±0.01	0.85±0.03	
FBP+U-Net	34.07±1.01	0.97±0.01	0.87±0.02	35.09±1.07	0.97±0.01	0.89±0.03	36.14±1.13	0.98±0.01	0.93±0.01	36.65±1.20	0.98±0.01	0.95±0.01	
Proj2Proj	26.00±0.77	0.90±0.01	0.84±0.03	26.10±0.81	0.90±0.01	0.88±0.02	26.16±0.83	0.91±0.01	0.91±0.01	26.21±0.85	0.91±0.01	0.94±0.02	

The reconstruction of human CT image results were given in Figs. 4 and 5 with different noise levels. When the reconstruction quality of all methods was compared, a strong analogy was observed with those obtained from Shepp-Logan and ellipses dataset. In Fig. 4, streak artifact is observed in the image (i). Since FBP method is used as the initial reconstruction for our method, the structured artifact that arises from the initial reconstructor (FBP) is prolonged by the proposed network. The noise level also plays an important role in the formation of structural artifacts. The noise level in Fig. 4 is higher than the one in Fig. 5 which adversely affects the reconstruction quality of FBP the most. As a result, the image (i) reconstructed by Proj2Proj in Fig. 5 does not have similar artifacts.

Figure 4 Human CT image reconstruction results from 64-view with 33 dB SNR noise level: (A) ground truth, (B) FBP, (C) SART, (D) SART+TV, (E) SART+BM3D ( σ=0.35), (F) SART+BM3D ( σ=0.20), (G) DIP+TV, (H) FBP+U-Net, (I) Proj2Proj trained on human CT dataset.

Image source credit: Yang et al. (2018).

Figure 5 Human CT image reconstruction results from 64-view 37 dB SNR noise level: (A) ground truth, (B) FBP, (C) SART, (D) SART+TV, (E) SART+BM3D ( σ=0.35), (F) SART+BM3D ( σ=0.20), (G) DIP+TV, (H) FBP+U-Net, (I) Proj2Proj trained on human CT dataset.

Image source credit: Yang et al. (2018).

Figures 6 and 7 show the 1-D intensity profiles passing through the dashed line from the reconstructed images which enables us to see how smoothly the intensities change. As the numerical and visual evaluation suffers from poor results, FBP, iterative, and regularized reconstruction methods are not given in the comparisons except for SART+BM3D with σ=0.20. The black line represents the 1-D intensity of ground truth and other methods while the red line is a copy of ground truth to show how well the reconstruction methods fit over the ground truth. In Fig. 6, SART+BM3D shows an over-smoothing through the line while DIP+TV shows undesired spikes along the same line. Proj2Proj and FBP+U-Net have similar line intensity profiles but Proj2Proj has slightly sharper edges than FBP+U-Net. In Fig. 7, a similar analogy is observed with SART+BM3D and DIP+TV methods. The ground truth image in Fig. 7 is the one reconstructed with normal-dose taken from the dataset and thus it consists of small minor spikes throughout the line which can be considered as the noise by its nature. FBP+U-Net here smooths out these spikes and creates a smoother image however in our proposed Proj2Proj method, it is evident that the small spikes are also preserved.

Figure 6 The 1-D profiles of the reconstructions from left to right: ground truth, SART+BM3D ( σ=0.20), DIP+TV, FBP+U-Net, proposed Proj2Proj method.

Image source credit: Yang et al. (2018).

Figure 7 The 1-D profiles of the reconstructions from left to right: ground truth, SART+BM3D ( σ=0.20), DIP+TV, FBP+U-Net, proposed Proj2Proj method.

Image source credit: Yang et al. (2018).

Another parameter that should be considered is the reconstruction time given in Table 3. Although the training times are quite long for the self-supervised method, the reconstruction times on the trained neural networks are very short thanks to modern computer technology.

Table 3 The execution times of the reconstructions in seconds.

FBP	SART	SART+TV	SART+BM3D	DIP+TV	FBP+U-Net	Proj2Proj	
0.38	34.16	36.91	41.95	320.20	1.21	1.27	

Discussion

Widely used image quality measurement metrics, SSIM and PSNR, have some limitations such as smoothness-biased and pixel-to-pixel difference measurement rather than focusing on perceptual similarity. Besides, in CT imaging, the reference images obtained from normal-dose CT scans contain noise and artifacts by its nature. The supervised method learns a mapping from low-dose to normal-dose images including all these imperfections as well. This might lead the supervised method to generate very high PSNR and SSIM scores even though it may not give the best reconstruction quality. Recently a new approach was proposed (Zhang et al., 2018) to measure the similarity between two images which outperformed traditional metrics when we compare them with human preference. For CT reconstruction problem, the perceptual difference is more important than pixel-to-pixel differences. Therefore LPIPS metrics were also added to the quantitative results.

One of the challenges of our proposed training method is the convergence of the network weights to an identity function. To overcome this problem, our method uses the Jth invariance method proposed in the Noise2Self (Batson & Royer, 2019) study, which performs a pixel-based denoising assuming the independence of inter-pixel noise. In a low-dose CT image, noise cannot be modeled independently on a pixel-by-pixel basis due to the back projection operator. Therefore, the Jth invariance method was applied in the projection domain. When the self-supervision principle was also tested in the image domain, the network converged to an identity function, which led the network to produce a result almost identical to the output of the FBP operator.

One of the questions that may arise is why noise reduction is done in the image domain. Although there are studies that eliminate noise in the sinogram domain (Lee et al., 2019; Anirudh et al., 2018), CNN neural networks are generally designed for natural images, and sinogram images have different characteristics from natural images. Besides, DIP study (Ulyanov, Vedaldi & Lempitsky, 2018) proposes an unsupervised method for image domain inverse problems by exploiting the structure of Deep CNNs.

Another discussion regarding the implementation of the proposed method might be choosing FBP as the initial reconstruction method. FBP, the most conventional reconstruction method, is used in FBP+U-Net (Jin et al., 2017), and a very similar architecture with FBP+U-Net was chosen for a fair comparison. On the other hand, iterative or regularized iterative methods for the initial reconstruction might help to obtain improved results. Among various supervised low-dose CT reconstruction methods, FBP+U-Net was found one of the best and fairest candidates to be used in the comparisons considering its network architecture and reconstruction approach.

Clinical applicability, which is one of the biggest factors in measuring the success of a reconstruction method, increases when all details can be reconstructed and structural details can be easily examined by radiologists. In our study, the deep lesion (Yan et al., 2017) dataset was selected as medical CT data and we aimed to predict the lesion detection performance of reconstruction methods. In addition to the superiority of the Proj2Proj method in quantitative analysis, the fact that it is clearly more successful than the other methods in the recovery of fine details, which shows its clinical applicability potential.

Conclusion

In this study, it was shown that low-dose CT reconstruction problem can be tackled by defining a training scheme to use low-dose projections as their own training targets even if the low-dose/normal-dose pairs of large datasets are not available. Even though traditional metrics might favor other methods depending on test setups and noise levels, considering detailed visual assessments including 1-D profiles and a novel deep CNN based quality assessment metric which evaluates the clinical usability of the results the best compared to the other metrics, Proj2Proj method is the most favorable one among all others including a well-known supervised method, FBP+U-net. It has great potential in other medical imaging problems where the same limitations exist and the assumptions used in this study are valid. As an advanced stage of this study, the method can be adapted to other imaging modalities.

Additional Information and Declarations

Competing Interests

Author Contributions

Data Availability

The authors declare that they have no competing interests.

Mehmet Ozan Unal conceived and designed the experiments, performed the experiments, analyzed the data, performed the computation work, prepared figures and/or tables, and approved the final draft.

Metin Ertas analyzed the data, prepared figures and/or tables, authored or reviewed drafts of the article, and approved the final draft.

Isa Yildirim conceived and designed the experiments, authored or reviewed drafts of the article, and approved the final draft.

The following information was supplied regarding data availability:

The DeepLesion dataset is available at NIH Box (Ronald Summers): https://nihcc.app.box.com/v/DeepLesion.

The simulated data, Ellipses dataset, is available at GitHub (dival/dival/datasets/ellipses_dataset.py): https://github.com/jleuschn/dival/tree/master/dival/datasets.

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
