# Peer review of "Proj2Proj: self-supervised low-dose CT reconstruction"

_PeerJ Computer Science, doi:10.7717/peerj-cs.1849_

## Round 0.1 · original submission · Major Revisions

Based on the referee reports, I recommend a major revision of the manuscript. The author should improve the manuscript, taking carefully into account the comments of the reviewers in the reports, and resubmit the paper.

**Language Note:** The review process has identified that the English language must be improved. PeerJ can provide language editing services - please contact us at copyediting@peerj.com for pricing (be sure to provide your manuscript number and title). Alternatively, you should make your own arrangements to improve the language quality and provide details in your response letter. – PeerJ Staff

Reviewer 1 ·

Basic reporting

This manuscript presents a self-supervised “projection to projection” method for low-dose CT. While the paper is easy to read, I have a few concerns with the current manuscript.

Experimental design

1. Even the FBP images are not terrible in this study. We would like to see the results of proposed method with even lower dose levels (sparser views, noisier projections), so that we know the full potential of the proposed approach.

2. "parallel beam projections were generated using Radon transform; image resolution was selected as 512x512; detector resolution was selected as 512x1 pixels". These settings are too ideal and far from reality. It is unclear whether this method is practical for clinical use.

3. The authors compared the performance of the proposed method with several iterative reconstruction approaches. It is also important to include state-of-the-art deep learning approaches as a comparison.

Validity of the findings

1. Is the trained mapping f(theta) different for different datasets because of the noise/sparsity level? Could the authors shed more light on this aspect?

2. In figure 3g and figure 4g (not figure 5g), why are there many streak and patchy artifacts? Especially, figure 4g, Projection2Projection trained on human CT dataset, the resulting image is still worse than figure 4f.

Additional comments

n/a

Cite this review as

Reviewer 2 ·

Basic reporting

Here are some comments that I think the authors should address to improve the paper.

-Why did you just compare the method with the analytical methods? Briefly explain the motivation for selecting the following algorithms for comparison. FBP, SART, SART+TV, SART+BM3D. The method can also compared with a state-of-the-art unsupervised or supervised deep neural network-based method.
- Please clarify how data are split for training, validation and testing for reproducibility of the algorithm
- Do you think the proposed network is hyperparameter dependend considering the clinical usability ?
- What was your criterion while selecting the certain perturbation ?
- Writing should be standardized and checked.
- Captions of all figures should be rearranged correctly.

Experimental design

no comment

Validity of the findings

no comment

Cite this review as

Reviewer 3 ·

Basic reporting

You addressed an important topic but need language corrections (proofreading should be done). Authors need attention to be paid to the clarity of expression and readability. (especially the use of references within the text; please check the citation format of the journal. eg. Line 50-51 etc.)

The first time you use the term, you can acronym it and continue using the acronym thereafter. Also, please do not introduce an acronym unless you will use it a minimum of three or four times. Unfortunately, you used many more acronyms, but some of them were not explained(eg. FBP etc.), and some were used once in the text, especially in the related work section (you do not need to give the acronym if you will not use it again).

What are the keywords of the study?

Experimental design

Please replace the paragraph at the end of the Introduction that explains your new approach (the first sentence is not a good place to start). Also discuss your work in comparison to your preliminary work published at the conference. What's the difference or what did you add? What is the important contribution of this study to the literature? From what I've examined, it appears to be a repeat of a similar study; Therefore, please emphasize its differences and innovation.

Also, the Figure 1 you used is also available in your other publication, so it would be better to reference it.

What is the size of each dataset, is it sufficient for CNN algorithms?
How do you split the training and reconstruction part or can you use k-folding?
The comparison metrics are PSNR and SSSIM. What are these and how can they be calculated?
Additionally, the iteration rate and learning rates you use are determined by where. After making a comparison and getting results with different values, it should be stated whether the most optimal values are, if so.

Validity of the findings

The paper identify clearly implications for research, and these implications reasonably good consistent with the findings and conclusions of the paper. Although the results are presented, I still do not feel fully convinced. In discussion part, need support for the arguments and claims.

Additional comments

In my opinion, since the continuous use of the pronoun "we" in the written language causes informal writing, passive uses will make the text more readable so that it is more formal.

Cite this review as

Reviewer 4 ·

Basic reporting

See the attachment.

Experimental design

See the attachment.

Validity of the findings

See the attachment.

Additional comments

See the attachment.

Annotated reviews are not available for download in order to protect the identity of reviewers who chose to remain anonymous.
Cite this review as

---

## Round 0.2 · accepted · Accept

Author has addressed reviewer comments properly. Thus I recommend publication of the manuscript.

Reviewer 1 ·

Basic reporting

The authors have addressed my previous comments. The revised manuscript is much improved and ready for publication.

Experimental design

In Table I and Table II, we see different levels of noise from 30dB to 40dB. Could you add details and describe clearly how the different noises were simulated?

"The projections were contaminated with noise between 30-40 dB." This is too brief for other researchers to reproduce the experiments.

Validity of the findings

n/a

Additional comments

This manuscript is ready for publication, pending minor revision.

Cite this review as

Reviewer 2 ·

Basic reporting

The authors made all my revision requests. There is no additional revision.

Experimental design

The authors made all my revision requests. There is no additional revision.

Validity of the findings

The authors made all my revision requests. There is no additional revision.

Additional comments

no comment

Cite this review as

Reviewer 3 ·

Basic reporting

The answers given to the criticisms made in the previous review are sufficient and appropriate. I have no other comment.

Experimental design

The answers given to the criticisms made in the previous review are sufficient and appropriate. I have no other comment.

Validity of the findings

The answers given to the criticisms made in the previous review are sufficient and appropriate. I have no other comment.

Cite this review as

Reviewer 4 ·

Basic reporting

Accept

Experimental design

Good

Validity of the findings

Yes

Additional comments

NA

Cite this review as